# Accountability-Aware Design of Voice User Interfaces for Home Appliances

Soheil Kianzad[1]*, Yelim Kim[1]*, Julia A.B. Lindsay[1]*
Yue Huang[2], Julian Benavides Benavides[2], Rock Leung[3], Karon MacLean[1] †

[1]Department of Computer Science, University of British Columbia, Vancouver, BC Canada
[2]Department of Electrical and Computer Engineering, University of British Columbia, Vancouver, BC Canada
[3]Samsung Research & Development Canada , Vancouver, BC Canada

## ABSTRACT

The availability of voice-user interfaces (VUIs) has grown dramatically in recent years. As more capable systems invite higher expectations, the conversational interactions that VUIs support introduce ambiguity in *accountability*: a user's or system's obligation or willingness to be responsible for the outcome of user-delegated tasks. When misconstrued, impact ranges from inconvenience to deadly harm. This project explores how users' accountability perceptions and expectations can be managed in voice interaction with smart home appliances. To explore links between degree of automation, system accountability and user satisfaction, we identified key design factors for VUI design through an exploratory study, articulated them in video prototypes of four new VUI mechanisms showing a user commanding an advanced appliance and encountering a problem, and deployed them in a second study. We found that participants were more satisfied with automated systems, but also saw them as more accountable. Results suggest that this perceived system accountability was reduced in mechanisms that justified their recommendations. Our findings motivate the development of transitional mechanisms which gradually provide less guidance as users learn to operate the system.

**Index Terms:** Human-centered computing—Interface design prototyping; Auditory feedback; User Interface design;

## 1 INTRODUCTION

Advances in artificial intelligence (AI) are changing how users interact with software agents. AI-infused systems vary in the level of automation they present to users: they can recommend options, make decisions, communicate with other agents, and adapt to their environments [6, 62]. A *Voice User Interface* (VUI) is a type of user interface that relies on speech recognition to communicate with users, usually with a conversational style [15] that resembles natural verbal intercourse, rather than manual clicking or typing.

Assistant-type VUIs are growing in popularity on personal devices. The majority of Americans own a smartphone (81%) or tablet (52%), which today come equipped with Siri, Google Assistant, or equivalent VUIs [61]. 1 billion devices worldwide running Windows 10 [43] provide access to Cortana, Microsoft's voice assistant. Access is different from use, but devices that exclusively accept voice input, *e.g.*, Amazon Echo and Google Home are on the rise: over 100 million Alexa-enabled units had been sold as of January 2019 [9]. It is clear that many consumers are newly choosing and trying voice interaction in their everyday life [2]. With this prevalence, we can go beyond 1st-order traits of this modality (hands-free,

natural language) to examine factors such as social consequences.

VUI technologies are often used for requests for information or to trigger applications [7]. Failure is annoying but not dire. However, as voice recognition technology improves and systems are capable of greater automation, users hold VUI systems to human-like standards of behavior. During VUI conversations with smart devices, users seem to anticipate accountability along similar lines as they would with humans; *e.g.*, Porcheron et al. describe users expecting appropriate responses, both verbal and actionable, from Alexa and express dismay when these are not provided. If the system makes an utterance, they react as they would to a human utterance [52].

However, AI systems that enable automation typically work under uncertainty, balancing false-negative and false-positive errors with potentially confusing and disruptive results [6]. Impact widens as standalone systems become platforms that control many home technologies through the Internet of Things (IoT) [7, 36, 45]. With Google Home, users can adjust lighting and set the thermostat [33], but also interact with systems invoking larger consequences. Smart washing machines can ruin clothes; personal assistant devices can spend money online. A semi-autonomous car can crash and kill in a moment of ambiguity over who is in charge.

So, what happens in the case of a bad outcome? Does the user hold the system responsible, or themselves?

*Accountability* can be defined as an entity's obligation or willingness to take ownership or ultimate responsibility for the outcome of a task, including one that has been delegated [3]. It is fundamental to how people conceptualize their actions and react to outcomes in a social context, by considering risk, uncertainty, and trust in taking or delegating ownership of outcomes [13]. Both a societal and individual concept, accountability differs subtly from broad definitions of responsibility: it is possible to be responsible (in charge) without being accountable if you take action but not ownership of the results.

Can interaction design mediate this balance, when it is important that the user retain accountability for a delegated task? We focus on **perceived accountability (hereafter accountability)**, which varies with user experience and expectations as well as situation (*e.g.*, degree of automation actually available), and therefore can vary by instance [62]. These factors impact a user's perception of system capability [18]. Because of these interlinked perceptions, we posit that through design we can manage user perception of a system's automation, and influence their notion of accountability.

*Research Questions:* We consider two questions in the context of VUI interaction with advanced home appliances:

**RQ1:** What design factors impact user perception of system accountability?

**RQ2:** How does automation influence perceived accountability and user satisfaction? Can interface design mediate this influence?

*Approach:* An invitation from industry colleagues to investigate user experience with voice-controlled smart appliances led us to consider VUIs in terms of user types, social roles, privacy and value added. In a first exploratory study, *accountability perception* emerged as

---
*These authors contributed equally.

†maclean@cs.ubc.ca

an important and understudied factor. We used insights from this exploration to propose primary *design factors* that can influence accountability during the user's interaction with the system (RQ1).

To go deeper, we studied more carefully how varying the level of system automation could influence user perception of system accountability. Our work is in the tradition of HCI community-proposed guidelines and recommendations for interaction with AI-infused systems. Early work by Norman [48] and Höök [30] sets out guidelines for avoiding unfavourable actions during interaction with intelligent systems, aiming to safeguard the final outcome by managing autonomy and requiring verification. Horvitz proposed a mixed-initiative method to balance automation with direct manipulation by users [29]. While these works discuss cautionary actions to avoid potential problems, their impact on users' perception of accountability in case of failure is unexplored.

We chose voice-controlled laundry machines as our focus application domain because modern washing machines can require more engagement than appliances like refrigerators or toasters, and have a plethora of complex settings that can cause confusion and errors. This complexity opens possibilities for guiding accountability perception, in a dialogue-type interaction. It is also apparently recognized in current industry activity, with Samsung and LG both developing washing machines with voice assistants.

We constructed video prototypes [68] featuring four different scenarios of VUI interaction for washing clothes, as design probes to provoke open dialogues with participants about accountability for each interaction scenario.[1] Set in a home environment, the VUI mechanisms vary in level of automation and instruction and as a group enable participants to compare degrees of accountability delegation.

Through in-person interviews and online questionnaires, we obtained participants' rankings of accountability and their satisfaction of each mechanism in light of the task failure illustrated. We analyzed this data in relation to the represented system automation.

***In contribution, we:***

- Propose primary design factors of accountability in voice user interactions with complex technology: *task complexity*, *command ambiguity* and *user classification*.
- Demonstrate the ability to *direct accountability perception* through VUI design, for the case of a smart appliance.
- Provide initial insights on the relationship between user satisfaction and system automation which can inform more generalizable examination.

## 2 RELATED WORK

### 2.1 Automation, Interaction & Accountability

***Internet of Things and Smart Home Appliances:*** IoT technology connects users and environmentally-embedded "smart" objects, from individual gadgets (like smartphones and smartwatches) [31], communal appliances (smart speakers, thermostats, vacuum robots), to semi-autonomous systems and sensor networks) [42, 67]. The exploding number of IoT devices and complexity of controlling them can negatively impact user attitudes [49]. We note that while considerable smart technology is available today, our study scenarios imbue VUIs with slightly futuristic decision-making ability.

***Intelligent user interfaces (IUI):*** In addition to sensor capacity and the IoT, some smart home appliances benefit from embedded IUIs. IUIs simplify interactions through AI capabilities such as adaptation [32] and the ability to respond to natural commands and queries with apparent social intelligence. Suchman's discussion of situated action highlights the need for context-dependent responses in HCI [63]. Although situated-action models result in versatile

and conversational systems, this approach is based on probabilistic behaviour which is prone to unexpected errors.

***Task delegation to AI:*** Studies in AI-human interaction focused primarily on systems capabilities such as reliability and cost of decision (*e.g.*, [51]). Considering human preferences and perceptions, Lubras *et al.* summarize the literature in shared control between humans and AI, and propose four factors for AI-task delegation: risk, trust, motivation, and difficulty. Emphasizing human perception, their research supports the human-in-the-loop design and low preference for automation [41]. However, the user's accountability perception generally does not appear in these studies.

***Explainable-accountable AI:*** For both usability and ethical reasons, algorithmically derived decisions should be *explainable* [19, 34, 58]. Systems utilizing them should provide accounts of their behaviour and inform users about sensor information, resulting decisions, and likely consequences [8, 20]. Some argue for policies on automated decision making [22], and a few governments have established regulations that require AI systems to provide users with explanations about any algorithmic decisions [24]. Explainable AI enables human to make sense of the machine learning models and understand the rationales behind AI decisions. Abdul *et al.*reviewed over 12,000 papers from diverse communities on trends in explainable-accountable AI [4]. They highlight a lack of HCI research on practical, usable, and effective AI solutions. Other groups have found that classic UX design principles may be insufficient for AI-infused products, and we need to develop guidelines specifically for human-AI interaction [6] — a motive of the present work.

***Automation and accountability:*** Previous work on accountability mainly focused on social accountability among humans, and how people justify or explain their judgments [10, 65]. Some, however, investigate accountability during collaborative decision making of a human and an intelligent agent [18, 46, 57]. Skitka *et al.*show that holding users accountable for their performance reduces automation bias (too much trust in automated decision makers), improves performance and reduces errors [60]. Suchman shows how the agency attributed to a human or a machine is constructed during an interaction [63]. Others have investigated how users negotiate and interpret their agency while interacting with VUIs [38, 59].

However, no study has yet shown how to *direct* the perception of accountability through design. Accountability research in HCI goes beyond usability and deserves significantly more attention from intelligent user interface designers, including VUI designers.

***Control Capabilities***: Building on works from Dourish, Button and Suchman [12, 20, 63], Boos *et al.* propose that users feel they have control over a system based on its "**control capabilities**" – specifically, when it is *transparent*, *predictable* and can be *influenced* [10]. The authors further suggest that users who feel in control of an interaction are more likely to consider themselves accountable for the outcome. We aim to determine whether users can identify subtle differences in control capabilities, and whether that affects the accountability of the system.

### 2.2 Voice-User Interfaces

We use Porcheron's definition of a VUI, which specifies interfaces that rely primarily on voice, such as Amazon's Alexa or the Google Assistant [52]. They are always on and can be accessed from room-level distances, which results in them being highly "embedded in the life of the home" compared to other technologies. The quality of their human-centered design is imperative.

The union of VUIs and smart home appliances is largely unexplored despite its promise; *e.g.*, IoT is one of the most frequent VUI command categories that users employ in their daily interactions with home assistance devices [7]. While we see this as a great opportunity to incorporate VUI into home appliance technology, we heed Dourish's advice to "take sociological insights into the heart of the process and the fabric of design" [21].

---

[1]Please check the supplementary materials for more information about the video prototypes and their scripts .

Our work is distinguished from past efforts in VUI use in every-day life [52] by moving beyond understanding users' perception of accountability and trying to direct it through design.

## 3 EXPLORATORY STUDY

While some design factors have been identified at the boundary of automation and human interaction (*e.g.*, trust, state learning, work-load, machine accuracy [50]), we needed specific insights for VUI semi-automated systems. To answer RQ1, we investigated user experience with VUI products relative to non-VUI-controlled but "smart" home products. We did this through interviews (n=10) and questionnaires (n=43), recruiting through social media (Facebook, Twitter). The results, briefly summarized here, motivated using VUIs and suggested where accountability matters most.

### 3.1 Methods

*Participants:* We targeted past purchasers of smart home appliances. Of 43 questionnaire respondents (20/21/2 F/M/unreported), age range was 25-55 years, from Canada, USA, Colombia, UK, China and Australia. All did or had owned smart home appliances.

*Questions:* Participants reflected on their experiences with smart home appliances and voice-command technology, compared voice with other input modalities, and considered VUI integration for two hypothetical smart systems: lighting, and a washing machine. They were asked to imagine the functions these systems might fulfill through VUI commands, and explain any concerns.

### 3.2 Results

Accountability figured strongly in responses, emerging as a an under-explored design lever. Results further exposed three factors framing the situational impact of accountability: *User Classification*, *Task Complexity* and *Command Ambiguity*.

*Motivations and De-motivations for VUIs:* Our participants appre-ciated VUI speed, convenience (particularly hands-free use), multi-tasking, shallow learning curve, and natural language. In contrast to human conversations, they wished to minimize interactions. When describing envisioned VUI smart home appliances, we heard that they needed a "*machine that can decide for [itself]*." [P8] How-ever, they were concerned about unreliability, hesitating to use VUI for complex tasks with irreversible outcomes, and concerned about misinterpretation, likely from prior experience.

*Factor I – User Classification:* We observed *primary* users, in charge of choice and maintenance, and *secondary* users reliant on the primary. Consistently, [56] notes that home technological man-agement is not evenly distributed by gender or across the household.

*Factor II – Task Complexity:* Participants categorized home appli-ances mainly by interface complexity, not underlying technology. We thus subsequently focused on home appliances with more com-plicated UIs and non-trivial consequences of failure.

*Factor III – Command Ambiguity:* While positive overall, partici-pants cited examples of concern which we categorize as naive access, hidden functionality and open-ended requests. Natural language is inherently ambiguous, requiring the system to make assumptions and decisions, as with human-human interactions. In so doing, account-ability can be delegated – important to recognize should something go wrong. We seek design factors that influence this delegation.

## 4 FRAMEWORK

### 4.1 Accountability via "Control Capabilities"

We explored how VUIs can affect accountability using Boos *et al.*'s theoretical framework of Control Capabilities (Section 2.1), which is based on the premise that "*in order to answer account-ability demands [...], certain requirements of control need to be fulfille*d" [10]. We framed our experimental study around an exten-sion of this proposition and framework, seeking to verify or disprove

it. To the *Transparency* and *Predictability* dimensions proposed by Boos et al. [12] we added *Reliability* because of its prominence in our exploratory study.

*Transparency:* Transparency can be achieved through executing clear and understandable actions. Several studies recommend im-proving transparency by providing explanations about the behaviour of AI-empowered systems [28, 35, 40, 53].

*Predictability:* Predictability can be obtained by producing desired and anticipated outcomes. Human-AI guidelines suggest two points where interactions with an AI should be shaped: *over time* and *when wrong*. They advise that during an interaction, a system should con-vey updates to users regarding future consequences of the system's behaviour, and support invocation of requests as needed [6].

*Reliability (added):* Reliability can be achieved through delivery of desired outcomes based on given explanations. A well-studied construct in automated systems, trust is crucial in long-term adop-tion [27] and key for voice interaction [11]. To invoke trust, we chose *reliability*: the quality of performing the correct actions.

### 4.2 User Satisfaction as a Metric

User satisfaction with automation generally improves with reduced cognitive effort. However, a system can avoid accountability by requesting detail, *e.g.*, by providing choices or asking for confir-mation. This increases user involvement, at the potential cost of satisfaction. Measuring user satisfaction as well as accountability perception indicates how well that balance is achieved.

We defined this metric based on principles of measuring cus-tomer satisfaction level [39, 55], then designed a questionnaire to assess emotional satisfaction by asking about: (a) overall quality (Attitudinal), (b) the extent user's needs are fulfilled (Affective and Cognitive) (c) users' feelings (Affective and Cognitive) [1].

## 5 PRIMARY STUDY: METHODS

### 5.1 Overview and Hypotheses

Since humans can manage accountability in their conversations, we surmise that designers should be able to enable this in human-machine interaction. We hypothesize a correlation between automa-tion level (from fully machine-controlled to fully user-controlled), and system accountability. Our goal was to focus on how the level of automation influences both accountability and user satisfaction, which eventually inform the design of interactive systems.

We conducted a controlled experiment with 15 survey respon-dents, of which 8 were also interviewed. Participants watched a series of video sketches [68] showing four levels of automation, where an individual uses a VUI with a smart washing machine for both simple and complex tasks. In every video, the washing machine fails to fulfill the user's expectations, since accountability is relevant primarily when the system fails.

Participants were instructed to imagine themselves as the user. We surveyed their perceptions of the washing machine's accountability for each VUI mechanism to obtain quantitative data, followed by open dialogues on accountability, satisfaction and general thoughts about each scenario.

Our hypotheses address the joint effect of system automation and task complexity on system accountability with the future goal of employing them in balance. We anticipated that:

**H1:** Increasing users' involvement in decision-making (thereby decreasing system automation) will reduce their perception of system accountability, particularly for high-complexity tasks.

**H2:** As we increasingly automate task decision-making, user satis-faction will increase.

If these hypotheses are correct, then system automation creates a trade-off between system accountability and user satisfaction. This work explores user perceptions surrounding this trade-off in the

context of a VUI interaction. We also investigate the effects of task complexity on accountability and user satisfaction.

## 5.2 VUI Accountability-Directing Mechanisms

To direct users' accountability perception, we conceptualized four VUI mechanisms representing levels of system accountability, based on guidelines for a progression of automation in AI systems [29, 30, 48]: *automation*, *recommendation*, *instruction*, and *command*. We created video dialogues by following highly cited guidelines [5, 17, 25, 26]. The levels differ primarily in the degree of direct manipulation, automation and information conveyed, and method of information delivery. We captured the mechanisms in walkthrough-style video prototypes for use in the study task.

*Automation* presents a straightforward workflow: the user requests an outcome and the VUI notifies them of the action to be taken: *e.g.*, after the user states they would like to wash their clothes as quickly as possible, the machine chooses to execute a quick wash cycle. Because of the system's take-charge approach, we anticipate that users will regard this largely as a delegation of accountability to the system. This accountability delegation comes into play in failure cases. For example, in the above case of the quick wash cycle, if the clothes are not cleaned as effectively as a normal wash.

*Recommendation* provides options based on the user's description of the clothes. For example, after the user describes his clothes as *colored*, *made of no special material*, and *medium load*, the mechanism provides two suggestions with different temperatures and spin speeds. The user selects one. We posit that here, the system is accountable for the *quality of recommendations*, but the user who makes the choice is ultimately responsible for the *outcome*.

*Instruction* provides the most information. It guides users in examining their clothes, and based on description and requirements, explains multiple washing suggestions. For example, after suggesting an *extra rinse*, the machine gives a detailed justification. If the user feels they have enough information, they can stop by saying 'Stop, I choose the first suggestion'. Here, we expect the user to hold the system accountable only for instruction accuracy.

Although users have equivalent choices in Instruction and Recommendation, they differ in presentation of the choices. We expect this to be reflected in the Control Capabilities measures of Transparency, Reliability and Predictability.

*Command* enables the user to set the washing cycle without any information from the machine. Users simply state their requirements instead of pushing buttons. This implies that the user knows what she wants. With this mechanism, we do not expect the user to hold the system accountable for the outcome.

## 5.3 Design and Variables

As we investigate influence of control capabilities (2.1), rather than measures of the system's *actual* controllability by the user, we are looking for how a designer can increase a *user's sense* of control. In this study, such insights appear from the principles we used in mechanism designs (5.2); in the choice of outcome measures (5.5); and the application of the measures, confirming whether the design manipulations were impactful.

The study itself uses within-subject 2x4 design, with independent variables of *complexity level* (low/high, described below), and *VUI mechanism* (4 mechanisms). For each complexity level, we counterbalanced the order of VUI mechanisms.

*Task Complexity:* With our exploratory study revealing the importance of task complexity and failure consequences on accountability perception, we varied task complexity for insight into H1 (whether design, via increased user involvement in decisions, can mediate perception of system accountability in high-complexity tasks).

*Low complexity* – Routine laundry, common in a household.

*High complexity* – A job involving special material (wool), extra requirements (stained fabric), and non-standard functions.

In our videos, for the low complexity condition the system attempted to remove mud from clothing; for high complexity, to remove wine stains from a valuable sweater. For all conditions, the washing machine failed to completely clean the clothes.

## 5.4 Procedure

We recruited homeowners with purchasing power for home appliances, using social media advertising and referral of participants (similar to [37, 44]), necessary due to the inclusion criteria and lack of participant compensation. We recruited a subset of survey participants to be interviewed in-person, immediately post-survey. The survey took an average of 30 minutes, and the follow-up interviews 20–45 minutes, average 27 minutes.

Survey participants answered a demographic questions and watched eight video prototypes: four distinct mechanisms, each performing a high- and low-complexity task. Videos were labelled by numerical order of appearance, counterbalanced by participant. Participants were then asked to rank the mechanisms by "how accountable each one was for the failed laundry task". Then, they scored each mechanism for Control Capabilities of Transparency, Predictability and Reliability and User Satisfaction. We asked the interviewee participant subset to verbally explain their responses.

## 5.5 Data Collected

The pre-video questionnaire (28 questions) collected participants' demographic information and past experience with non-smart washing machines, including whether they tended to hold non-smart washing machines "responsible for failed laundry tasks"). The post-video questionnaire collected participants' ratings for participants' satisfaction and Control Capabilities for each video (*i.e.*, VUI mechanism), while the interviews collected qualitative justifications of participants' survey responses.

*Accountability Ranking:* After watching each mechanism fail to complete the washing task, participants ranked them (1 (most) to 4 (least) accountable, tie not allowed). Ranking facilitated direct comparisons between short lists of items [47].

*Control Capabilities:* We sought participant opinions on Control Capabilities (Transparency, Predictability and Reliability) for each VUI mechanism as presented in the video prototypes. As with [14, 23], they scored each mechanism for Control Capabilities using a slider on a [0-100] point scale.

*User Satisfaction:* Again with a [0-100] point scale and a slider, we asked participants to respond to three questions:
- How easy would it be to use the voice-assisted system?
- How confusing was the voice-assisted system?
- How satisfied would you be with this interaction?

## 5.6 Analysis

*Perceived Accountability Rankings:* We performed Friedman tests (widely used for ranked data [47, 54]) on the mechanisms' accountability ratings to identify any correlation between accountability perception and system automation (which varied with the VUI mechanism in each video), for each level of task complexity (high or low). In post-hoc analysis, we used Bonferroni correction of confidence intervals to compare accountability rankings by VUI mechanism. For all statistical results, we report significance at $\alpha = 0.05$.

*Control Capabilities & User Satisfaction:* We analyzed each set of [0-100] scores with a repeated-measures ANOVA. Due to a violation of sphericity, we report Greenhouse-Geisser results. Post-hoc analysis included a Bonferroni alpha adjustment. User Satisfaction scores were taken by averaging participant responses to the three questions listed in 5.5, which provided a broad depiction of ease of use, clarity and interaction experience.

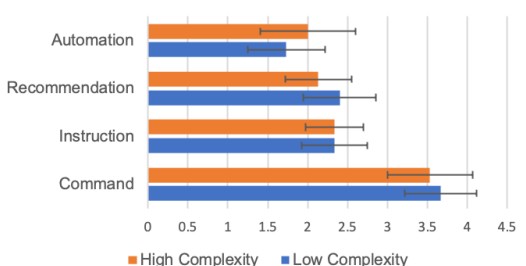

Figure 1: Average responsibility rankings of experiment's VUI mechanisms by task complexity, for question "*How accountable (responsible) is the system if something goes wrong?*" Rank 1 (greatest) to 4. Error bars are standard error of mean.

*Interviews:* We used Braun and Clark's approach for thematic analysis [16]. In repeated passes, two investigators conducted open coding. Afterwards, two other team members checked the coding and brought disagreements to the full team for resolution. This division provided a broader perspective, deepened our understanding and generated multiple discussions around each theme.

## 6 RESULTS

We recruited 15 survey participants (10 male, 5 female, age distribution M = 34.97, SD = 7.86). Of these, we interviewed 8 (3 male, 5 female). Participants were from various ethnic backgrounds but all lived in North America at the time of recruitment.

### 6.1 Quantitative Results: Questionnaires

*Pre-Questionnaire Data:* We surveyed participants on their past experiences with household technology, including their technology roles within their households. In our exploratory study, we identified two role classifications. *Primary* users are enthusiastic about initial setup and ongoing maintenance of home technology; we designated other users as *secondary*. All respondents indicated they had purchasing power within their households.

$\sim 73\%$ of participants reported enthusiasm in exploring new features on their smart home appliances, and took responsibility for configuring home technology. This suggests that the majority of the participants were primary technology users based on our definition.

As an assessment of how participants related the notion "accountability" to washing machines, we asked where they placed the blame when a non-smart washing machine damaged their clothes. 60% had had that experience and "mostly" or "completely" blamed their washing machine. This seems to dispel a notion that perceived accountability skews towards self in such situations.

*Perceived Accountability Rankings:* Figure 1 shows participants' rankings of mechanism accountability for the portrayed outcome.

Friedman tests on task complexity found differences in accountability ranking to be statistically significant across VUI mechanisms.

For low-complexity tasks, $\chi^2(3, N = 15) = 18.28, p < 0.001*$; for high-complexity, $\chi^2(3, N = 15) = 13.32, p = 0.004*$. Post-hoc analysis (Table 1) with Bonferroni correction of confidence interval found that for both high and low complexity tasks, *Command* had significantly lower accountability than *Automation* and *Recommendation*. For low-complexity tasks, the *Command* mechanism had significantly lower accountability than *Instruction*.

*Control Capability scores:* We analyzed participant scores for each mechanism in the Control Capability (CC) dimensions of *Transparency*, *Predictability* and *Reliability*. Differences between CC scores for the *Recommendation* and *Instruction* mechanisms (Figure 2) suggest that option delivery impacts experience of control over the interaction. For example, though *Recommendation* and

Table 1: Relative VUI Mechanism Accountability (Bonferroni-adjusted).

| Mechanisms Compared | Low-Complexity | High-Complexity |
|---|---|---|
| Command-Automation | $z = -4.10, p < 0.001*$ | $z = -3.25, p = .007*$ |
| Command-Recommendation | $z = -2.97, p = 0.018*$ | $z = -2.97, p = .018*$ |
| Command-Instruction | $z = -2.83, p = 0.02*$ | $z = -2.546, p = .065$ |
| Automation-Recommendation | $z = -1.131, p = 1.0$ | $z = -.283, p = 1.0$ |
| Automation-Instruction | $z = -1.273, p = 1.0$ | $z = -.707, p = 1.0$ |
| Recommendation-Instruction | $z = -.141, p = 1.0$ | $z = -.424, p = 1.0$ |

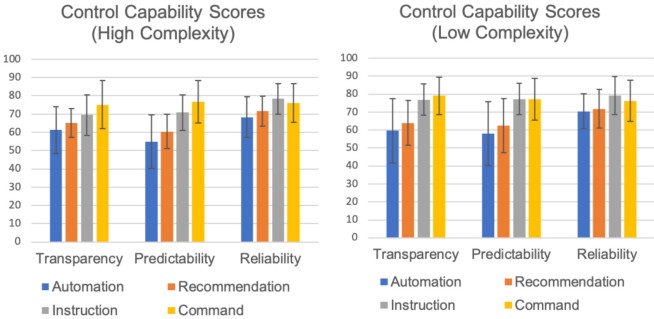

Figure 2: Participant scores (1-100) on control capabilities of VUI mechanisms (15 samples / bar). Error bars are standard error of the mean.

*Instruction* offer similar choices to users, *Recommendation* was consistently seen as less transparent and predictable, as well as more accountable in high complexity tasks.

Figure 2 reports average ratings for CC dimensions by VUI mechanism; it shows a trend suggesting that increased automation is linked to reduced perceived transparency and predictability. We found statistical significance only for predictability for the high complexity task. However, in our post-hoc test with Bonferroni alpha adjustment, we were not able to find any statistical significance between specific mechanisms for predictability.

*Accountability and user satisfaction for low-complexity task:* The trend of the average satisfaction scores in Figure 3 suggests that participants preferred the *Automation* mechanism to those requiring more user involvement, for both low and high-complexity tasks. Participants also reported higher satisfaction with *Instruction* and *Command* for high task complexity. However, we did not find statistical significance for either tasks (High Complexity: F(2.075, 29.05), p=0.576; Low Complexity: F(1.885, 26.391), p=0.258).

### 6.2 Qualitative Results: Interviews

Eight questionnaire participants (4 female), selected through snowball sampling [66], were interviewed (Section 5.5). All were adults living with others who self-identified as primary or secondary users of home appliances, meaning they had purchasing power for home appliances in their households. No compensation was provided.

In the following we organize our analysis of the interview transcripts as laid out in Section 4. Mechanism names here replace the numerical labels that participants used to refer to the videos.

Table 2: ANOVA results for Control Capabilities and User Satisfaction

| Ctrl Capability | Low Complexity | High Complexity |
|---|---|---|
| Transparency | F(1.848, 25.877), p=1 | F(1.887, 26.423), p=0.314 |
| Predictability | F(1.514, 21.193), p=0.079 | F(1.94, 27.164), p=0.028* |
| Reliability | F(1.917, 26.834), p=0.305 | F(2.044, 28.616), p=0.275 |
| Satisfaction | F(1.885, 26.391), p=0.258 | F(2.075,29.05), p=0.576 |

*Greenhouse-Geisser results are presented due to the violation of Sphericity. A post-hoc test with Bonferroni alpha adjustment was not significant for predictability.*

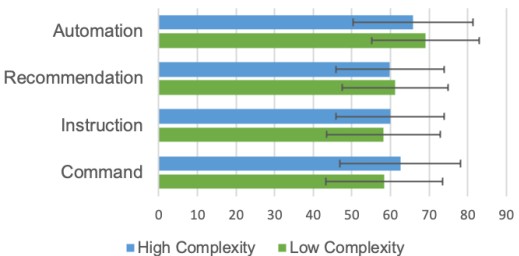

Figure 3: Average user satisfaction score (1-100) for low complexity and high complexity tasks, by VUI mechanism. Error bars: standard error of the mean.

When asked to revisit their ranking of accountability across the VUI mechanisms, the majority of participants identified *Automation* as the most accountable. However, this was not unanimous: P3 suggested all mechanisms were "completely accountable", and P8 found the *Recommendation* mechanism most accountable. These individual variations further justify the use of Control Capabilities to identify design factors that contribute to accountability.

***Automation is seen as most accountable:*** *Automation* was deemed by the majority of participants (both primary and secondary) as the most accountable, because the machine gives minimal information and selects the washing cycle by itself. "*Not given a choice*" [P4] and "*machine do[es] whatever... it feels the best*" [P7] are reasons that participants ranked the automation mechanism as most accountable.

> '*I think in [Automation], the machine should take the most responsibility since it makes all the decisions. . .'[P2]*

***Shared accountability:*** *Recommendation* and *Instruction* were viewed as generating a sense of shared accountability, by giving suggestions for participants to choose from. Participants are still accountable for the final decision: if "*something goes wrong, it should be your fault instead of the machine*" [P5]. However, "*maybe the user will blame the machine for giving [...] the wrong recommendation*" [P6] in the case of an error. Some suggested that since the system "*understands the situation [...] it's more accountable*" [P7].

***Control Capability Dimensions:***
*A. Transparency* – Participants generally agreed that all mechanisms were transparent enough for them to understand the interactions. As in Figure 3, most of participants rated and mentioned *Command* and *Instruction* as "more transparent" [P3, P4, P5, P7]. Some viewed *Command* as "*the most transparent*" [P4], since the user has complete control in this traditional method of doing laundry.

> '*...To me transparency relates to the extent to which they use or understand what's going on within the machine, so in video number one that I mentioned a machine is pretty much making the selections on behalf of the user [...] so the user doesn't really know what's going on. Whereas [ Automation] for instructions, the machine just told me what the user is saying so the user one hundred percent still all the time what's going on. . .'[P7]*

*Instruction* was also viewed as transparent since it provides detailed information and participants understand the procedure.

> '*...the [instruction-based] with lots of details. Also, it's transparent. Although it's a bit annoying but it is transparent. . .'[P4]*

Some responses seem to indicate that the participants found the amount of information excessive. Additionally, some participants found *Automation* clearer since the interaction process was less complex and the "*machine takes care of everything*" [P7].

*B. Predictability* – Participants tended to consider *Instruction* as predictable since it is "*the most specific*" [P6] and it "*give[s] explanations*" [P1]. Participants claimed that they "*trust it most"* [P1] and described it as an expert guide:

> '*It's so smart, the machine acts as a teacher to teach you like, uh, what should you do? I don't have to worry about anything. He just tells you everything. . .' [P5]*

The *Command* mechanism received high predictability ratings. Some participants suggested that the user in the video must have been familiar with the system already:

> '[Command based mechanism]... cause the user knows what he or she wants and maybe that's because he/she has already tried it before. Then for the [instruction based mechanism] because you have all the descriptions of the options the results would also be predictable' . [P7]

*C. Reliability* – Participants also tended to view *Instruction* as the most reliable for both simple and complicated tasks, since they gather the most information and it "*seems to know a lot*" [P1, P2].

> '*[...] if something goes wrong, it should be your fault instead of the machine. Because the machine let you know all the consequences before.'[P5]*

> *Interviewer: Why do you think that the instruction based mechanism is more reliable? . . . " "... I believe that the machine knows what it is doing because it has all the information about the laundry process that I don't"[P4]*

***Satisfaction:*** Participants' satisfaction with the interaction depended on both the VUI mechanism and task complexity. The "*concise [..] and very quick"* [P3] *Automation* mechanism was the most satisfying for some, especially for routine tasks because "*you don't have to think about what you're doing*" [P6].

For complex, critical or high-stakes washing tasks such as "*really expensive clothes*" [P6], the "*detailed instructions*" [P3] of instruction-based mechanisms were considered more satisfying. Participants appreciated additional information when the perceived cost of error was high (*e.g.*, damaging expensive clothes).

> '*... it depends on what you're trying to wash like if I'm gonna wash really expensive clothes that I cannot mess up. Uh, then I would have done the third one because whatever I don't know what to do, it tells me exactly what's the stuff to take separate the clothes and all that stuff.'[P6]*

## 7 DISCUSSION

This study demonstrates a difference in accountability between our designed mechanisms. The results of both qualitative and quantitative analysis supports our first hypothesis of a positive relationship between system automation and accountability. This relationship is well represented by P2's interview response that *"...the machine should take the most responsibility since it makes all the decisions."*. A similar trend has been reported by Sheridan *et al.*: *"... individuals using the [automated] system may feel that the machine is in complete control, disclaiming personal accountability for any error or performance degradation*" [57].

Though our results showed that *Automation* takes the highest accountability, two participants provided insights on other potential design factors that affect accountability. P5 argued that *Instruction* should be accountable when the system fails because users "*wasted*" their time listening to its verbose instructions. P6 indicated the importance of claims about the system: *"Actually that really depends on what the machine says it can do, you know if it says like 'I'm*

*gonna be able to distinguish colour clothes from regular clothes and I'm not gonna mess up.' and it messes up then it's the machine's fault.''* Setting realistic expectations about a system's abilities may help manage its accountability.

Our results echoed Suchman's prediction that automation can lead to shared accountability between humans and machines [64], and further that level of automation impact perceived accountability.

Users did not experience the washing machines firsthand, a limitation imposed by the state of technology. However, participants empathized with the common experience of doing laundry sufficiently to report a projected level of satisfaction with the interaction. They showed no difficulty in bridging the gap between experienced and imagined scenarios, making comments such as *"there is common ground between me and the machine"* [P3].

Our results suggest that task complexity does influence user satisfaction. Our qualitative analysis made it clear that for the high complexity laundry task, participants were more willing to ask for guidance and more likely to include the VUI agent in the decision-making process compared to the low-complexity task. However, they might prefer *Command* or *Automation* once they became comfortable with the system. Multiple users expressed the desire to transition to command-based systems once they had learned about the washing machine's hidden functionality through the instruction-based or recommendation-based mechanisms. This key finding motivates VUIs during naive access, which could be invaluable for secondary users of home appliances.

The qualitative results also support our quantitative analysis outcomes. Some participants stated that only experienced users who found the washing machine predictable would use the command-based mechanism. This may have contributed to an inflated predictability rating for the command-based mechanism. Though results suggest *Automation* is perceived as the most accountable, and *Command* the least, it is difficult to make a conclusive judgment on shared accountability for *Recommendation* and *Instruction*. Each of these mechanisms was scored differently in one Control Capability; however, the difference was not significant.

## 7.1 Implications for Design

We encapsulate these findings in a set of recommendations for VUI design of complex technology. Drawn from a study of prospective interactions with one class of technology, further investigation is required to broadly generalize them, but they form a positive first step that can be built upon.

1. Accountability-aware design must consider context, specifically *task complexity*, *type of user(s)* and *ambiguity* of the interactions.

2. Automation has opposing effects on *accountability* and *user satisfaction*. A highly automated system may be satisfying to use, but in case of failure, users are more likely to find it blameworthy. Designers should consider this trade-off in the unique context of their product and its anticipated use.

3. User perceptions of *Reliability*, *Transparency* and *Predictability* depend both on available choices, and how those choices are presented, particularly for high complexity tasks. Designers should consider providing justification for system-presented choices, especially for high-stakes tasks. Doing so may help manage users' perception of the system's accountability.

4. Results suggest that detailed instruction- and recommendation-based mechanisms improve learnability, but could eventually be too repetitive. Designers should consider transitional mechanisms, in which system operation gradually provides less explanation and becomes more automated. This type of transitional mechanism bypasses the automation/satisfaction trade-off described above in Implication 2.

## 8 CONCLUSIONS AND FUTURE WORK

We investigated the concept of accountability in home appliance VUIs. We examined automation level as a parameter that could impact accountability delegation, by designing and studying four mechanisms which varied automation and user involvement in decision-making, in simple and complex tasks.

Our primary study sought to characterize differences between these mechanisms. We found our use of video prototypes a successful basis for initial discussions on design concepts, providing non-trivial insights.

Qualitative and quantitative results support our first hypothesis of a positive relationship between automation and accountability, which held whether for both high and low complexity tasks.

Concerning our second hypothesis (that system automation increases user satisfaction), the quantitative result (N=15) was not statistically significant, but trended towards users preferring the most automated system. Interviews consistently supported H2 in that increased user involvement reduced satisfaction. This creates a dilemma for designers of automated systems, who must minimize users' cognitive load without saddling the system with complete accountability for errors.

Automation non-withstanding, control capabilities of reliability, transparency and predictability offer potential to manage system accountability. This became evident in comparing the *Instruction* mechanism with the *Recommendation* mechanism. Both mechanisms are equally automated, with the user responsible for selecting washing settings. The difference between them is that the *Instruction* mechanism provides justification for the options it presents, whereas the *Recommendation* mechanism does not. The latter was consistently seen as more accountable, less transparent and less predictable. The manner in which options were presented to the user, *ie.* by providing justifications or not, affected users' perceptions of the systems' accountability, despite equal degrees of automation.

We found participants more receptive to instructions and recommendations when they were concerned about the outcome of a process, and during their first use of a system. We recommend that VUI designers implement guided interfaces as well as command-based ones. This gives users the freedom to transition from guided use to command-based use without leaving the system (and its designers) accountable for mistakes.

***Future Work:*** From this foundation, we recommend next steps.

*Sample Size* – Our study size was appropriate for this early stage of investigation, revealing clear trends supporting the possibility of directing perceptions of accountability in users to support greater investment (more realistic study approaches) in this idea. However, increasing the size and diversity of even this exploratory approach might provide higher power of statistical tests and more significant quantitative insights.

*Mechanism design* – We examined four distinct mechanisms in isolation. As suggested in Section 7.1, we propose a mechanism that adjusts its automation as the user becomes more familiar with the device. The benefits of such a mechanism would need to be confirmed in a longitudinal study.

*Metrics* – User satisfaction is a volatile metric. In this study it is especially so because participants did not interact with a physical prototype. To minimize this limitation on user empathy, we assessed common moderate failure outcomes instead of complete failures (*i.e.*, a stained rather than a destroyed shirt). We will have more realistic results when users can reflect on their satisfaction level by observing the laundry process outcome on their own clothes.

*Realistic results* – A functional VUI system and, separately, a machine that truly enacts its instructions would advance the reality of the participants' experience and make their responses more reliable. A real system would succeed more often than fail, as opposed to our scenarios which aimed to make use of a short study session. When

studied longitudinally within real homes in actual use, we can follow the development of trust and familiarity over time.

## 9  ACKNOWLEDGMENTS

This work was supported by the University of British Columbia's Designing For People (DFP) program. We appreciate the participants who helped us with this study, and thank faculty and student members of DFP for their valuable feedback.

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
