# OpenReview forum: "Accountability-Aware Design of Voice User Interfaces for Home Appliances"
_graphicsinterface.org/Graphics_Interface/2021/Conference — GI 2021_

### Official Review · AnonReviewer3 · 2021-01-12
**Nice study and overall well written paper**

**Rating:** 8
**Confidence:** 2

**Review:**

This paper describes a study into the perceived accountability of voice user interfaces. Participants watched videos of interactions with a washing machine in four conditions varying in how much control the user delegates to the machine. This is crossed with complexity in a high and low condition. The main takeaway, in my perspective, is that as more control is delegated to the washing machine, users' also tend to "blame" it more for negative outcomes.

I think overall the paper is well done and presents a solid piece of research. The investigate aspect of accountability is timely and interesting, the study setup good, and I also see no issues with the analysis.

One thing that could be improved is the clarity around accountability and how it was measured. The definition given is that "Accountability can be defined as who (e.g., user or system) is obligated or willing to be responsible or accountable for the satisfactory execution of a task, including one that has been delegated". So accountability is when someone/something is accountable. That's oddly circular and I'm wondering whether that can be expressed differently. Also, in that statement responsible and accountable read like the exchangeable things, yet just below we learn that they do differ. In the main study, accountability is then elicited by asking participants to rank the four conditions from 1 (most accountable) to 4 (least accountable). This does lead to a slightly hard to read presentation of the results though. For example, in Figure 1, less is more accountable. If accountability is shown, it would be more intuitive if higher values corresponded to a higher sense of accountability.

But I don't think there are major issues and hence I am recommending acceptance.

---

### Official Review · AnonReviewer1 · 2021-01-12

**Rating:** 6
**Confidence:** 5

**Review:**

This paper presents a study to understand user perspectives on the perceived accountability of washing machine voice user interfaces (VUIs). The researchers deployed an initial survey to elicit perspectives on the possibilities and concerns about VUIs used in the home setting. Next, they created video prototypes illustrating various errors that might occur when washing machines use VUIs and used them in interviews with 15 participants to elicit their perspectives on system accountability.

The area of VUI design and especially their home applications are relevant areas of research in HCI and the current paper asks important questions about how users perceive errors and who should be accountable for them when considering these systems. The paper nicely grounds its analysis on previous work on accountability in HCI. The methods are, for the most part, clearly described and the findings can be used to inform accountability aspects of design in future VUIs for smart home appliances.

Despite its strengths, the paper has several important shortcomings. First, while the findings mostly focus on smart washing machines in the study sections, they are broadly generalized to home all automation systems in the Discussion and Implications for Design.  This is problematic since the findings (both survey and interviews) are still exploratory and in the absence of multiple studies with different types of appliances, it is difficult to generalize the results to the perceived accountability of VUIs broadly. I believe this issue can be addressed effectively by revising different parts of the paper, especially the Introduction (i.e., framing) and Discussion/Implications for Design and specifying that the current findings are specific to VUIs for smart washing machines (with the possibility of some of them generalizing to other systems that need to be verified in the future). I would also mention the specific application under study earlier (in the Abstract or the Introduction at the latest) than where it is currently described (methods). Second, given the relatively small number of participants in the main study (15) I was surprised that the quantitative analysis was more detailed that the qualitative analysis, and that the latter was only limited to 8 participants. Given the exploratory nature of the study and that its focus is on users’ perception, I would have liked to see more detailed qualitative data and analysis.  Also, I was wondering what type of thematic analysis the authors conducted? Finally, I found the Implications for Design section one of the weaker sections because the results are quite general and it is hard to imagine how they may be applied to actual design scenarios. I recommend removing and incorporating with the Discussion or Conclusion. I end with a minor structural suggestion: the first paragraph of section 6.2 can move to methodology and the second paragraph reduced to one sentence about the naming convention.

---

### Official Review · AnonReviewer2 · 2021-01-13
**Well done study but weak contribution**

**Rating:** 5
**Confidence:** 4

**Review:**

The paper "Accountability-Aware Design of Voice User Interfaces for Home Appliances" presents an exploratory study on the perception of accountability of automated systems. The authors explore what affects the perception of system accountability and the relationship to user satisfaction. The general domain of the paper is voice-controlled interfaces and the application domain is voice control of laundry machines. The study is based on showing participants (15) video prototypes of voice-based interactions with laundry machines and assessing their perception of accountability through questionnaires and interviews. Results show that the higher the level of automation the higher the level of perceived accountability. The authors recommend designers of voice user interfaces to give freedom in transitioning between guided use (more automation) and command-based use (more direct instruction).

The paper addresses a timely problem as voice-based user interfaces are becoming more and more common, yet still a source of frustration and error. The paper is well written, well structured, and generally easy to follow. There is good coverage of related work and the theoretical framework for accountability is nicely presented. The methodology of the paper is sound and the authors nicely combine questionnaire and interview data.

I also have some concerns with the paper:
 - The design of the video prototypes seems to be very important to the participants' perception of accountability. However, the design decisions that were made making the video prototypes are not detailed, and what the video prototype shows is not shown (e.g., in images) or explained to the reader. Basically, the reader is kept completely in the dark about one of the most important parts of the study.
 - I am not sure what I have learned from reading this paper. The conclusion could be summarized with "The more responsibility we give to the machine, the more accountable we perceive it to be for the outcome". I don't see this as surprising at all.

Overall, the paper presents a relevant and nicely executed study, however, it fails to deliver a convincing contribution and glosses over the design and description of the video prototypes that are central to the paper. Therefore, I lean towards rejecting the paper.

---

### Meta-Review · Area_Chair1 · 2021-01-15

**Recommendation:** Accept
**Confidence:** 5

**Metareview:**

The reviews of this paper are mixed but leaning towards acceptance. All reviewers find that the topic is timely and important, the paper covers related work nicely, and that the methodology is sound. However, the reviewers also identify a number of shortcomings:
 - The quantitative analysis is more detailed than the qualitative analysis even though the paper presents an exploratory study
 - Implications for design are weak and too general
 - The design of the video prototypes is not motivated and described
 - The concept of accountability and how it was measured needs more clarity


I am leaning towards accepting this paper, however, there are a number of concerns raised by the reviewers that the authors will have to address:
 - Clarify the contribution of this particular study and tone down the generalization to automation systems in general
 - Briefly clarify how the video prototypes were designed and give a concise description of them
 - Strengthen the description of accountability based on the input from the reviewers
 - Carefully the reviewers' suggestions for improvements in the final version of the paper

---

### Decision · Program_Chairs · 2021-01-16

Accept